# Semi-Supervised Pipeline for Autonomous Annotation of SARS-CoV-2 Genomes

**DOI:** 10.3390/v13122426

**Published:** 2021-12-03

**Authors:** Kristen L. Beck, Edward Seabolt, Akshay Agarwal, Gowri Nayar, Simone Bianco, Harsha Krishnareddy, Timothy A. Ngo, Mark Kunitomi, Vandana Mukherjee, James H. Kaufman

**Affiliations:** 1AI and Cognitive Software, IBM Almaden Research Center, San Jose, CA 95120, USA; akshay.agarwal1@ibm.com (A.A.); gowri.nayar@ibm.com (G.N.); sbianco@us.ibm.com (S.B.); hvkrishn@us.ibm.com (H.K.); timothy.ngo@ibm.com (T.A.N.); Mark.kunitomi@ibm.com (M.K.); vandana@us.ibm.com (V.M.); jhkauf@us.ibm.com (J.H.K.); 2NSF Center for Cellular Construction, San Francisco, CA 94158, USA

**Keywords:** genomics, SARS-CoV-2, COVID-19, genome annotation, bioinformatics, gene prediction, protein prediction, protein domain, computational biology

## Abstract

SARS-CoV-2 genomic sequencing efforts have scaled dramatically to address the current global pandemic and aid public health. However, autonomous genome annotation of SARS-CoV-2 genes, proteins, and domains is not readily accomplished by existing methods and results in missing or incorrect sequences. To overcome this limitation, we developed a novel semi-supervised pipeline for automated gene, protein, and functional domain annotation of SARS-CoV-2 genomes that differentiates itself by not relying on the use of a single reference genome and by overcoming atypical genomic traits that challenge traditional bioinformatic methods. We analyzed an initial corpus of 66,000 SARS-CoV-2 genome sequences collected from labs across the world using our method and identified the comprehensive set of known proteins with 98.5% set membership accuracy and 99.1% accuracy in length prediction, compared to proteome references, including Replicase polyprotein 1ab (with its transcriptional slippage site). Compared to other published tools, such as Prokka (base) and VAPiD, we yielded a 6.4- and 1.8-fold increase in protein annotations. Our method generated 13,000,000 gene, protein, and domain sequences—some conserved across time and geography and others representing emerging variants. We observed 3362 non-redundant sequences per protein on average within this corpus and described key D614G and N501Y variants spatiotemporally in the initial genome corpus. For spike glycoprotein domains, we achieved greater than 97.9% sequence identity to references and characterized receptor binding domain variants. We further demonstrated the robustness and extensibility of our method on an additional 4000 variant diverse genomes containing all named variants of concern and interest as of August 2021. In this cohort, we successfully identified all keystone spike glycoprotein mutations in our predicted protein sequences with greater than 99% accuracy as well as demonstrating high accuracy of the protein and domain annotations. This work comprehensively presents the molecular targets to refine biomedical interventions for SARS-CoV-2 with a scalable, high-accuracy method to analyze newly sequenced infections as they arise.

## 1. Introduction

The ongoing SARS-CoV-2 pandemic has undoubtedly shaped our lives as one of the most significant global health challenges of the 21st century. However, unlike previous pandemics, we now have sequencing technology with tremendous throughput to analyze the genomic content of SARS-CoV-2. As labs around the world sequence isolates from infected individuals, we can track and characterize the viral genome evolution for near real-time surveillance.

The first sequenced SARS-CoV-2 genome [1] was submitted to NCBI on 17 January 2020 and has become the accepted reference standard commonly referred to as the Wuhan reference genome (NCBI RefSeq ID: NC_045512.2). Since that point, the sequencing of SARS-CoV-2 isolates has increased dramatically to tens of thousands of genomes a week. The SARS-CoV-2 genome is comprised of a 29,000 base pairs (bp) single-stranded RNA (38% GC content) with four structural proteins, two large polyproteins, which are cleaved to form non-structural proteins, and several accessory proteins [2,3]. There are two overlapping open reading frames responsible for Replicase polyprotein 1a (pp1a) and Replicase polyprotein 1ab (pp1ab), which yield the longest products from the genome and the majority of the non-structural proteins.

In comparison to other coronaviruses, SARS-CoV-2 differs phenotypically with its significant increase in transmissibility and asymptomatic or presymptomatic transmission, as well as genotypically from its polybasic furin cleavage site insertion in the S protein [4]. However, it maintains several other *Coronaviridae* traits, such as gene order consistency and transcriptional slippage [5]. The −1 programmed ribosomal frameshift responsible for transcriptional slippage has been observed to occur at the point where ORF1 (responsible for pp1a) continues as ORF1b (responsible for pp1ab) and is defined by an RNA signature marking the slippery site [6]. This phenomenon allows the virus to control the relative levels of its protein expression [6] and may be useful in therapeutic targeting to limit protein production. Additionally, this unique trait creates a challenge for traditional bioinformatic genome annotation programs, which assume that the more typical continuous 5′ to 3′ translation can be effectively used to form the correct protein sequence, which is not the case for SARS-CoV-2.

There are several viral genome annotation methods, such as VAPiD, Prokka, InterProScan, and others [7,8,9], that aim to provide autonomous (that is, no reference genome required) annotation of genes and proteins. Some of these tools have issued special releases to aid in annotating SARS-CoV-2 genomes. Yet, many of these tools are for more general purposes, do not provide sufficient accuracy with “off the shelf” use for SARS-CoV-2, and have not yet been applied at scale as the available SARS-CoV-2 sequence data grows. Additionally, several variants of SARS-CoV-2 genomes have emerged, including the D614G [10] variant, which appeared earlier in the pandemic, or the more recent B.1.1.7 (Alpha) or B.1.617.2 (Delta) variants [11,12], which represent the majority of new cases in the U.S.A. [13] and worldwide. The mutations defining these variants can present challenges for the complete automation of genome annotation, and this can be further exacerbated by the SARS-CoV-2 transcriptional slippage site.

As an alternative to an autonomous genome annotation method, alignments to the Wuhan reference genome [1] can be completed using tools such as NextStrain’s Augur [14], Bowtie2 [15], or UCSC SARS-CoV-2 genome browser [3]. This type of supervised analysis uses published gene coordinates to extract sequences from the query genome based on positional and sequence similarity to a reference genome. However, this creates a considerable dependency on a single reference genome, and reference-based assembly has disadvantages, as the resulting genome may be biased toward the used reference [16,17]. Since it is currently estimated that SARS-CoV-2 typically mutates approximately twice per month on any given transmission chain [18] and can be subject to recombination events [19], a reference-guided approach may face limitations as the virus continues to evolve or increase the rate at which it evolves.

In this work, we present a semi-supervised custom pipeline to annotate all genes, proteins, and functional domains for SARS-CoV-2. These sequences comprise the molecular targets to develop more accurate diagnostics, antivirals, and vaccines. Our semi-supervised method was applied to 66,905 SARS-CoV-2 genomes collected from NCBI GenBank [20] and GISAID [21]. This approach yielded nearly 13 million new molecular sequences and connections that can be accessed through the IBM Functional Genomics Platform, a tool made freely available to the COVID-19 global research community [22]. With this method, the expected proteins were correctly annotated at an average rate of 98.5% per protein across all genomes, and the average observed over expected protein length ratio was 99.1%. When evaluated against other tools, such as Prokka or VAPiD, we identified 6.4- and 1.8-fold increased protein annotations, respectively. Furthermore in a targeted analysis, we achieved greater than 97.9% sequence identity in spike glycoprotein domains and tracked the emergence of the D614G and N501Y spike glycoprotein variants over time and by region of exposure. Beyond this, we demonstrated the robustness and extensibility of our pipeline by analyzing an additional 4000 genomes sampled from current variants of concern and interest. From this set, we demonstrated high accuracy annotations in protein and domain sequences as well as achieving over 99% accuracy in identifying hallmark mutations in the spike glycoprotein associated with each variant.

The complete collection of SARS-CoV-2 genes, proteins, and functional domains continues to be updated and can be accessed via a web browser user interface or our developer toolkit (https://ibm.biz/functional-genomics, accessed on 23 March 2021). Ultimately, we present a comprehensive comparative analysis and data resource of 13,000,000 publicly available SARS-CoV-2 viral sequences with the aim of identifying potential targets to aid in vaccine, diagnostic, or therapeutic development.

## 2. Materials and Methods

We used a combination of state-of-the-art tools [8,9,22] and custom calibration tools to provide a semi-supervised genome annotation pipeline (Figure 1). We verified the accuracy of, and applied this method to 66,905 SARS-CoV-2 genomes to identify the gene, protein, and functional domain sequences contained within each genome. This collection was analyzed for accuracy and quality with regard to current known references. Protein variants are characterized as a function of time since the pandemic emerged and from a geographic perspective in an expansion of this pipeline to 4000 additional genomes.

### 2.1. Genome Data Retrieval and Quality Thresholds

SARS-CoV-2 genomes were retrieved from the Global Initiative for Sharing All Influenza Data (GISAID) [21] and NCBI GenBank [20] (retrieved 18 August 2020). A complete list of data sources, genome accessions, and acknowledgment of the submitting lab/author information where available is included in Appendix A. An md5 hash, unique identifier, was computed on each genome sequence (excluding headers) to track identical genome sequences. In preparation of genome annotation, two commonly used genome quality criteria and thresholds were assessed for their ability to yield a complete set of full length protein sequence annotations. Criteria A is defined as follows: genome length > 29,000 bp (only IUPAC characters allowed, gaps permitted), % unknown bases (Ns) < 1, “high coverage” flag indicated by GISAID defined as < 0.05% mutation density only in CDS, and no unverified indels in relation to all other genomes in GISAID. Criteria B is defined as follows: number of unknown bases <= 15, number of degenerate bases <= 50, number of gaps <= 2, and mutation density < 0.25. To evaluate genome quality criteria, all genomes were processed with our genome annotation pipeline (Figure 1) and their resulting protein sequences were evaluated. Protein length distributions as a function of this genome criteria were compared, using a two-sample Kolmogorov–Smirnov test (ks.test function in base R).

### 2.2. Gene and Protein Annotation

Specific refinements to our previously described genome annotation pipeline [22] were made to process SARS-CoV-2 genomes and yield gene, protein, and domain sequences. Key augmentations of Prokka v1.14.5 [8] were made for (1) improved unsupervised annotation of SARS-CoV-2 genomes and (2) the addition of custom-built supervised algorithms to improve the identification of specific proteins that were unable to be detected using the base implementation. The novel steps added by this work are indicated in blue in Figure 1 and a Docker container with these augmentations has been created for use at https://github.com/IBM/omxware-getting-started/tree/master/SARS-CoV-2_parser, accessed on 23 March 2021. The customizations for the gene and protein annotations are described briefly below and in further detail in Section A.1 and Section A.2.

To accommodate the nascent state of the SARS-CoV-2 reference data, the evidence codes required by Prokka to build the reference index used in gene and protein naming were loosened. Additional control surrounding the annotation mode, e.g., single or metagenome mode, were made to enable the single genome mode when the input genome was less than 100,000 bp in length to accommodate the length of SARS-CoV-2 (29,000 bp). A targeted search of intermediate gene coordinate data generated by Prodigal (within Prokka) for high sequence similarity to known protein references was added to augment genome annotations to include ORF9b, ORF10, and Envelope small membrane protein, which were otherwise missing from the annotations. For Replicase polyprotein 1ab, nucleotide sequences were harvested from intermediate gene coordinates and extended or fused to ensure full length gene products with correct start and stop codons. These sequences were verified to contain the slippery site [6] with nucleotide degeneracy and translated accordingly to accommodate to the known −1 programmed ribosomal frameshift occurring in SARS-CoV-2.

### 2.3. Protein Domain Annotation

Unique protein sequences were processed with InterProScan v5.48-83 [9] to identify domain sequences and InterPro (IPR) codes as previously described [22]. This version of InterProScan contains a number of InterPro, Gene Ontology and Pathway codes specific to the SARS-CoV-2 proteome and reference data. A full list of all available codes can be found at https://www.ebi.ac.uk/interpro/proteome/uniprot/UP000464024/, accessed on 23 March 2021.

### 2.4. Comparative Analysis

To compare our method against other published viral genome annotation tools, VAPiD (v1.2 with Python3) was run on a set of 100 randomly selected SARS-CoV-2 genomes above the quality control thresholds previously defined in Section 2.1, using the following parameters: reference (–r) NC_045512.2. Protein names and sequences were extracted from VAPiD output files using BioPython’s parser. Prokka version 1.14.5 [8] was run on this same set of genomes, using default parameters with –kingdom Viruses. The resulting protein sequences from each tool were compared for set membership per genome, protein sequence truncations and overall sequence similarity. Here, protein set membership accuracy is defined as ability to detect the set of expected proteins per genome.

To evaluate our method in the context of emerging and shifting SARS-CoV-2 variants, we expanded the initial genome corpus analyzed to include representation of all four variants of concern (VoC) and four variants of interest (VoI) defined by the Wold Health Organization and U.S. Centers for Disease Control as of 19 August 2021. This includes the following variants: B.1.1.7 Alpha, B.1.351 Beta, B.1.617.2 Delta, P.1 Gamma as well as B.1.525 Eta, B.1.526 Iota, B.1.617.1 Kappa, C.37 Lamda for VoC and VoI, respectively. For each variant listed, 500 genomes were selected randomly from the list of all SARS-CoV-2 genomes present in NCBI GenBank on 19 August 2021 for a total of 4000 additional diverse variant genomes. These genomes were subsequently annotated as described in Section 2.2 and Section 2.3 and analyzed for accuracy of gene, protein, domain, and mutation predictions.

Protein annotations were evaluated against the SARS-CoV-2 proteome reference sequences indicated in ViralZone, SIB Swiss Institute of Bioinformatics [23] for complete protein set membership per genome, sequence length, and sequence similarity to known references indicated in NCBI UniProt [24]. Set membership accuracy is the count of observations of a given protein for a set of genomes analyzed or in the case of domains, the set of domain sequences annotated for a given protein.

For domain accuracy comparative analysis, our predicted domains identified in the spike glycoprotein (S protein) were analyzed for set membership completeness against the expected InterPro domain architecture for UniProt reference sequence P0DTC2 (https://www.ebi.ac.uk/interpro/protein/reviewed/P0DTC2/, accessed on 12 February 2021). Additionally, where predicted domain sequences were assigned an IPR code (8146 unique domain sequences out of 9120 total domain sequences), the predicted domain sequence was compared against the reference sequence to yield a percent identity. Reference domain sequences were extracted from the S protein amino acid sequence (UniProt: P0DTC2) based on domain start and stop sites indicated at the link above. The amino acid percent identity was calculated with considerations for insertions, deletions, or substitutions.

For genome-to-genome and protein-variant comparisons (Section 3.1 and Section 3.4), genomeassociated metadata were retrieved from GISAID and processed for each analysis. For duplicated genome identification, an md5 of the genome sequence (excluding header) was completed as described in Section 2.1. The originating lab, date submitted, and host fields were used to further characterize candidate duplicate genome sequences. For the labeling of the spike protein variants, the date submitted and exposure region fields provided by GISAID for the containing genome were used to describe the time and geography of our predicted variants.

To identify spike protein mutations, the amino acid modifications for the eight variants of interest and concern were retrieved from Outbreak.info [25]. To select mutations that are hallmarks of a given variant, each mutation must be observed in at least 95% of the sequences analyzed by Outbreak.info for that variant. A multiple sequence alignment for spike protein sequences was completed per variant with the reference sequence included (UniProt RefSeq: P0DTC2) using Mafft [26] with default parameters. The mutated amino acid proportion was manually extracted from the consensus sequence with gaps excluded.

## 3. Results

Here, we present a novel semi-supervised pipeline (Figure 1) to annotate the gene, protein, and functional domain molecular targets from SARS-CoV-2 genomes. This pipeline builds on Prokka [8], a prokaryotic genome annotation tool, that is cited by nearly 7000 papers and was downloaded nearly 80,000 times from BioConda alone. We developed our pipeline to overcome key deficiencies of existing methods, such as incorrect Replicase polyprotein 1ab sequences, which are fully absent, falsely discontinuous, or artificially truncated in other methods. Additionally, ORF9b, ORF10, and Envelope small membrane protein were absent in annotations from other methods and indicated that a targeted search must be incorporated into our pipeline. Below, we demonstrate the resulting accuracy of our methods against known reference data and our advantage over other bioinformatic tools. We also evaluated the genome quality from data in public repositories and quantitatively evaluated commonly used quality criteria for their effects on the resulting annotations.

### 3.1. Assessment of SARS-CoV-2 Genome Quality in Multiple Data Sources

It is important to assess the quality of input genome assemblies before genome annotation in order to achieve accurate and high quality annotations. In this study, we analyzed a corpus of 66,905 SARS-CoV-2 genomes (Appendix A) deposited over the span of eight months from 108 countries into two key aggregate data sources: GISAID [21] and NCBI GenBank [20]. We observed an average of 0.0067% unknown bases (denoted as N per IUPAC definitions) per genome (unknown base range is 0–46.76%), and all genomes were observed to have fewer than 1% degenerate bases (Figure 2a). The presence of unknown bases can indicate insufficient genome coverage or other issues from genome assembly. Next, we aimed to identify criteria for inclusion of a SARS-CoV-2 genome assembly into our platform to ensure that the input data for molecular target identification were of the highest quality. Two commonly used criteria were evaluated for their effects on prediction of full length protein sequences. Briefly, Criteria A is more permissive and prioritizes the ratio of length vs. coverage, whereas Criteria B is more stringent and applies a higher penalty to the number of gaps (detailed definitions in Section 2.1). Here, “full length” is defined as a protein sequence length within 10% of the known UniProt protein reference sequence indicated in the SARS-CoV-2 proteome as defined in ViralZone [23,24]. Figure 2b demonstrates that Criteria A yielded the highest count of full length protein products (35,099 non-redundant protein sequences) while also effectively reducing the majority of truncated products (2197 non-redundant protein sequences). If applying the more stringent Criteria B, over 14,000 high quality protein products would be inadvertently removed. Based on this, we proceeded with applying the thresholds defined by Criteria A to the corpus of genomes analyzed in this work. From GISAID and GenBank genomes, 9.9% (out of 55,708) and 3.1% (out of 11,197) of the genomes fell below this criteria (Figure 2a) and thus, were removed from subsequent analysis and marked as low quality unless otherwise mentioned.

As part of our pre-processing steps for genomic data, we computed md5 hashes (a unique identifier) of all genome sequences to track duplicates. The rate of ‘duplicated’ SARS-CoV-2 genome sequences within a single data source or between data sources is indicated in Table 1. From the data at hand, it is unclear whether these duplicated genome sequences, as others have observed [27,28,29], are artifacts of data processing, e.g., due to alignments to a single reference genome, or are a result of sampling multiple patient infections within the same lineage. Of the 10,528 genomes that are duplicated within GISAID, we compared the metadata for each entry: 3953 are described with matching metadata entries in addition to identical full length genome sequences and therefore may be more likely to be data duplication artifacts. These potential duplication events are reported here but are not removed from subsequent analysis.

### 3.2. Quantification of Protein Sequence Prediction Accuracy

For an autonomous COVID-19 genome annotation pipeline to achieve clinical and biological relevance, it must accurately identify all known molecular targets within a genome. The SARS-CoV-2 proteome [2,23] is defined as having thirteen protein products, each with a corresponding gene sequence present in each genome. SARS-CoV-2 proteins are split into structural and non-structural groups, but all proteins are required for the virus to carry out its life cycle, which includes host cell invasion, replication, and transmission [30]. Using our gene and protein annotation method (Figure 1 and Section 2.2), we achieved an average per protein identification accuracy of 98.5 ± 2.9% across all genomes above the aforementioned quality thresholds. The number of observations per protein (Figure 3a) indicates that we were able to achieve complete or near-complete protein set membership for all genomes. Each protein is a translated gene sequence, and thus the equivalent gene identification accuracy is also achieved.

Furthermore, not only must the complete set of named genes and proteins be identified for accurate genome annotation, but the generated sequences must also be grounded in biological reality. Specifically, in silico predicted sequences should not be truncated with respect to the length of known references, and the mutational density must be low, considering the temporally recent emergence of SARS-CoV-2 and observed lower mutation rate, compared to other RNA viruses [18]. Using our semi-supervised gene and protein annotation method (Figure 1 and Section 2.2), we were able to identify full length protein products that, on a per protein basis, match the expected lengths of known reference sequences with an average observed/expected protein length value of 99.1% (Figure 3b). The distributions of our predicted and the expected protein sequence lengths are observed to be statistically similar by two-sample Kolmogorov–Smirnov test (D = 0.0071, *p* < 2.2 × 10^−16^) and are 8.75-fold more similar (D = 0.0617) than those predicted from genomes not passing our quality thresholds, i.e., low quality genomes.

In addition, certain gene and protein sequences required us to develop additional targeted methodological advances for identification (Figure 1 and Appendix A). Specifically, Replicase polyprotein 1ab (pp1ab) is the longest gene sequence within SARS-CoV-2, and its protein sequence is cleaved into 16 non-structural proteins [2]. It overlaps with Replicase polyprotein 1a, and during translation, undergoes −1 programmed ribosomal frame shift at what is known as a slippery site [6]. Both of these attributes make it more challenging to accurately identify with off-the-shelf in silico genome annotation methods. Specifically, in our benchmarking analysis (Section 3.3), base Prokka was unable to yield a single full length pp1ab sequence from any genome. Therefore, we implemented a semi-supervised method (Section A.2) to correct and extend the putative predicted gene coordinates for pp1ab and adjust the translation method to accommodate ribosomal frame shift, which is a problem that negatively affects other bioinformatic tools. Our algorithmic improvement yielded full length pp1ab sequences in all genomes and further achieved greater than 95% sequence identity to the reference pp1ab sequence (UniProt ID: P0DTD1) in over 99.15% of the variants we predicted (Figure 4).

Ultimately from this genome corpus, we were able to identify over 13 million gene, protein, and functional domain sequences in total (Table 2). Our system only stores each uniquely identified sequence once (distinct sequence), but maintains the relationship to its originating genome and to any connected sequences, e.g., gene, protein, or domain sequence, providing the total sequences identified (Table 2). The number of variants (distinct sequences) differs per molecular target across all bio-entities as well each variant’s frequency (cumulatively shown in the redundant count). Access to these data through the Functional Genomics Platform allows rapid investigation of these molecular targets.

### 3.3. Comparative Analysis of Genome Annotation Methods

With regard to pipeline accuracy, we evaluated our pipeline against VAPiD [7], which created a special release for annotating SARS-CoV-2 genomic data, and Prokka [8], a prokaryotic genome annotation tool for bacteria and viruses. From the same set of genomes, we contrasted the resulting protein annotations (Figure 5) in the context of set membership as well as in observed protein sequence length, compared to reference protein sequence length (expected). VAPiD and our method both achieved high accuracy with regard to truncated proteins, but our pipeline elicited more proteins in the highest accuracy category and 1.8-fold more protein annotations overall (Figure 5). Consistently, ORF9b and Protein 3a were missing from the VAPiD annotations. Prokka, on the other hand, did not yield any full length pp1ab protein sequences and generated a high amount of missing or truncated proteins, especially for Envelope small membrane protein, ORF9b, and ORF10 among others (Figure 5). Our method was able to identify 6.4-fold more protein products compared to base Prokka, and was able to generate full-length pp1ab products with high sequence identity to known UniProt references.

### 3.4. Investigation of Predicted SARS-CoV-2 Protein and Domain Sequences

As SARS-CoV-2 undergoes mutation events, a comprehensive catalog of variants is essential for developing molecular interventions with sufficient specificity and binding efficiency. In our initial genome corpus, we observed a median of 425 unique sequences with non-synonymous mutational differences per protein (the number of unique sequences range = 109–19,406) per protein. The S protein, which is involved in the invasion of human cells through interaction with ACE2 [31], is observed to have the highest number of variants among structural SARS-CoV-2 proteins (Figure 3a). Not surprisingly, the non-structural products of ORF1a and ORF1ab are also observed to have a higher amount of sequence variants, compared to other SARS-CoV-2 proteins (Figure 3a).

Since the S protein is key in the recognition of the host cellular receptor to initiate virus entry and host cell invasion and the target of multiple vaccines, antivirals, and diagnostics, we further examined its observed variants. By querying the S protein from the Functional Genomics Platform and accessing 4548 unique sequences, including metadata, such as collection date and exposure region, we observed two predominant S protein variants that shift in their cumulative frequency over time (Figure 6). Initially, an exact match to the reference spike glycoprotein sequence (green line, UniProt: P0DTC2) is observed most frequently. Then in mid-April 2020, the notable variant D614G (orange line) with now known increased infectivity, due to interaction with the ACE2 receptor [32,33], overtakes the ancestral reference sequence (green line) in its abundance, achieving fixation. Two other differing sequences are observed at lower abundance in this genome cohort and correspond to P1140X (olive line) and S2 cleavage product (pink line). Additionally, there are minor variants observed in fewer than 1% of the sequenced genomes. For example, we observed five protein sequences to contain the N501Y mutation from 13 genomes originating from Oceania (submitted 2 July 2020) and North America (submitted 1 June 2020). These variants are also observed to contain the D614G mutation, but are not present with the 69–70 deletion present in the B.1.1.7 variant of concern (U.K.) or B.1.351 E484K mutation (South Africa). Our observations are consistent with the current understanding of multiple introduction events causing the emergence of the N501Y variants [34]. Furthermore, since this mutation is observed in the B.1 and B.1.1 lineages which predate the current B.1.1.7 and B.1.351 variants, it further clarifies the current timing of mutational introduction points in the pandemic. Some of our observed sequence variants may be due to sampling limitations or data artifacts, e.g., sequencing or genome assembly error, but if a minor variant confers a selective advantage, its frequency could shift to become a more common variant as we have seen in recent months with the B.1.1.7 (Alpha), B.1.351 (Beta), and B.1.617.2 (Delta) variants [35]. The detection of these type of mutations is crucial to understanding viral evolution and sequence divergence. The identification of these mutations provides a jumping off point for further explorations in biochemical and structural experiments, which are part of common virological workstreams.

Related to understanding S protein function and interaction with host proteins, we further evaluated our predicted protein domains for their identification and sequence accuracy. Protein domains are sub-regions in the amino acid chain that are observed to determine biological activity, binding efficacy, and key structural components of a protein [36]. In the case of SARS-CoV-2 spike glycoprotein (S protein), these domains are responsible for the conformational change that occurs during host cell invasion and are also the targets of the current vaccines in circulation [32]. Therefore, the accurate identification of the presence and sequence of S protein domains is of the utmost importance when describing drivers of SARS-CoV-2 molecular characteristics. Specifically, we analyzed the set membership completeness of our predicted domains in S protein against the reference domain architecture defined by InterProScan for the S protein UniProt RefSeq (https://www.ebi.ac.uk/interpro/protein/reviewed/P0DTC2/, accessed on 12 February 2021). The S protein is canonically annotated to have nine domains necessary for its function (based on InterPro reference data at time of manuscript submission). From 5702 distinct spike protein sequences, our annotation method yielded an average 94.4 ± 4.4% domain set membership accuracy (Figure 7a) with minimal false positives (IPR043002 and IPR043614 observed in fewer than 0.1% of the proteins). For one of the expected domains (IPR043473), the domain architecture is split across two locations, and we correctly observed approximately twice as many counts as genomes. The corresponding count of distinct sequences for each domain is indicated in Figure 7a, and these domains were observed to have 679 unique sequences on average. The S1-subunit of the N-terminal domain (for SARS-CoV IPR044341 and Betacoronavirus IPR032500) was observed to have the highest count of non-synonymous variants. This domain is present on the side of the spike trimer and is involved in binding host receptors. Mutations in this region can affect differences in transmissibility and vaccine escape. To further evaluate the accuracy of our predictions in these key regions, we calculated the amino acid percent identity of each of our domain sequences against the reference domain sequences extracted from the spike glycoprotein reference sequence (UniProt: P0DTC2). All of our predicted domain sequences achieved greater than 98% median percent identity to the reference domain sequences (Figure 7b) and the very small percentage difference reflects the variation in S protein domain sequences from the ancestral strain and reference data. Together, this indicates the completeness of the annotation and correctness of the predicted domain sequences.

### 3.5. Methodological Robustness in Variant Diverse Genomes

Beyond the initial corpus used to develop this method, we aimed to assess the robustness of the method against the more recent variants that have emerged. A random sampling of SARS-CoV-2 genomes were retrieved from NCBI GenBank [20] across the eight named variants of interest and variants of concern by US CDC and WHO as of 19 August 2021. The resulting annotations were then evaluated for comprehensiveness and accuracy. In accordance with our previously described quality control steps, the variant-diverse genomes (500 randomly sampled genomes per variant) were assessed for quality, and 3079 high quality genomes out of 4000 total were observed. On average, 77.0% ± 9.0 s.d. passed the quality thresholds, with Beta and Kappa having the most low quality genomes (Figure 8a). Using this set of high quality variant-diverse genomes, we achieved 91.5–98.3% protein set membership accuracy across all variants, with the highest detection accuracy in the Delta variant (Figure 8a). Additionally, we were able to identify full length pp1ab sequences in over 99% of all genomes across all variants. Our spike protein domain annotations were observed to have an average 95.0% set membership accuracy. Beyond this, we assessed the ability of our annotation method to be able to identify key mutations in the spike glycoprotein. For this, we retrieved the set of mutations expected in at least 95% of sequences for each variant [25]. With overwhelming accuracy, we were able to identify all mutations of concern or interest across all variants and all other expected mutations for any of the variants with an average 99.8% accuracy (Figure 8b). This includes observations of deletions early in the S protein and differentiation between substitutions P681H, which is a mutation of interest, and P681R. This finding demonstrates the ability of our pipeline to scale with additional SARS-CoV-2 genomes across variants under the constraints of evolving sequence divergence. The high accuracy achieved in multiple attributes of our predicted protein and domain sequences demonstrates the robustness of this method as well as its scalability and extensibility for surveillance in the face of expanding viral evolution.

## 4. Discussion

Since the start of the SARS-CoV-2 global pandemic, there have been immense efforts globally to sequence with near real-time efficiency the viral genomes observed in infected patients. In order to capitalize on this large and growing corpus of data, high throughput computational methods must be developed for rapid, high accuracy analysis to deliver the molecular targets that are actually under evaluation for drug development, vaccine specificity, and diagnostic testing. The method described here provides one such avenue to accomplish this goal. The protein and domain data generated as part of this work provide these molecular targets in an efficient manner with very high accuracy across the entire SARS-CoV-2 proteome and for all genomes analyzed in this corpus, spanning multiple countries and lineages. Beyond this, our semi-supervised pipeline does not require the use of a single reference genome, which better allows the detection of novel or mutating gene, protein, and respective domain sequences as they emerge. The method described here was integrated with our Functional Genomics Platform and applied to hundreds of thousands of SARS-CoV-2 genomes. As the vaccination rates rise and the pandemic continues, this method can be used to rapidly monitor and track emerging protein variants across hosts and sampling niches, e.g., aerosol, wastewater, and surfaces, to inform disease understanding, vaccine specificity, and host protein binding affinity. Additionally as future work, further confirming the in silico predicted sequences, using a structural model will allow for refinement of the protein sequences and key domains to expand our understanding of interaction with host proteins, antivirals, or diagnostics. Overall, the data generated as part of this work provide a comprehensive set of protein and domain variants observed globally and support the research community as we continue to aim to understand and control the COVID-19 pandemic.

## Figures and Tables

**Figure 1 viruses-13-02426-f001:**
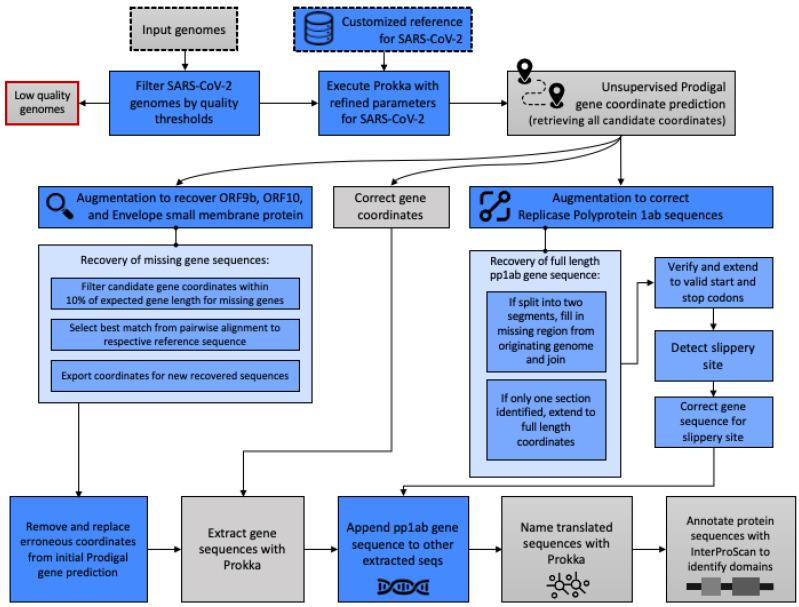
Pipeline schematic for semi-supervised identification of molecular targets. The data and analytic steps comprising our SARS-CoV-2 annotation pipeline described in Methods Section 2.1, Section 2.2, Section 2.3 are indicated. Input data are indicated with a dashed line and discarded data, e.g., genomes below quality thresholds, are indicated with a red line. Blue boxes indicate steps where our augmentation of base Prokka and Prodigal yield methodological advancements specific to the requirements of SARS-CoV-2 genomes.

**Figure 2 viruses-13-02426-f002:**
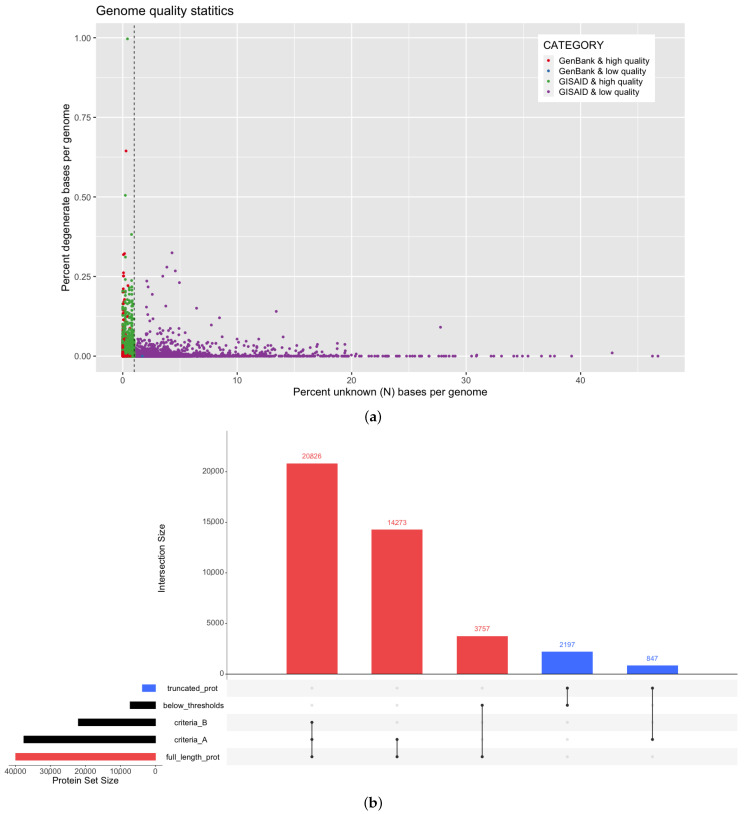
SARS-CoV-2 genome quality observations and their effect on annotation results. (**a**) The percent of unknown (N) and degenerate bases (as defined by IUPAC) are calculated as a function of total genome size for SARS-COV-2 genomes from two sources: NCBI GenBank and GISAID. The quality threshold of unknown bases is indicated with a dashed line. (**b**) Full length (red) or truncated (blue) protein products are indicated for genomes by quality criteria status: our selected criteria (Criteria A), more stringent criteria (Criteria B), or below quality thresholds. In this upset plot for each vertical bar, the categories defining the intersection are indicated with filled-in dots. For example, the first bar represents the number of proteins that pass both Criteria A and B and are also full length proteins, whereas the second bar represents the number of full length proteins that pass Criteria A but not Criteria B and are still full length. For criteria definitions, see Section 2.1.

**Figure 3 viruses-13-02426-f003:**
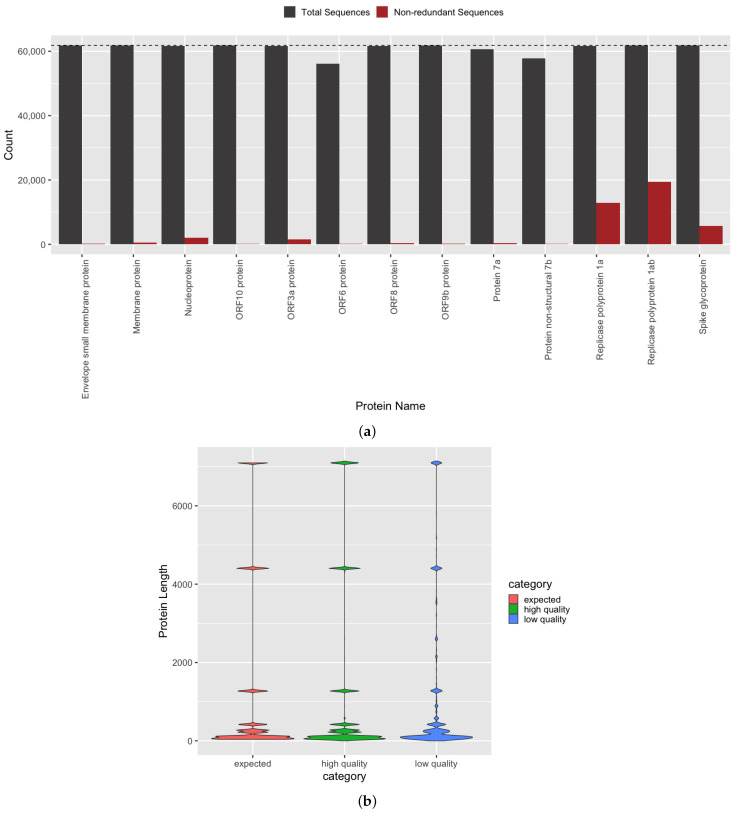
Protein annotation set membership and sequence accuracy. (**a**) Set membership of all (grey) and distinct (red) protein sequences identified in SARS-CoV-2 genomes above quality thresholds. Count indicates the number of times a given protein was observed in the entire high quality genome corpus (dashed line). (**b**) Protein size distribution of known protein references (expected, red) more closely matches our in silico predicted protein sequences from genomes above quality thresholds (high quality genomes, green), compared to products predicted from genomes below quality thresholds (low quality genomes, blue).

**Figure 4 viruses-13-02426-f004:**
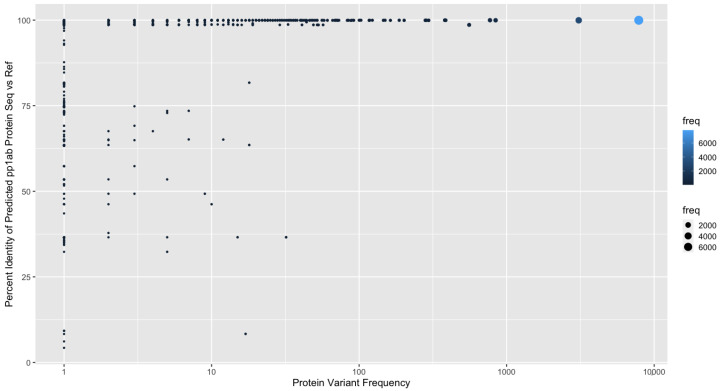
For Replicase polyprotein 1ab, the protein sequence variant frequency and protein sequence similarity of our predicted sequence to the known UniProt ID: P0DTD1 references is shown. The frequency of each protein variant is indicated by color and point size.

**Figure 5 viruses-13-02426-f005:**
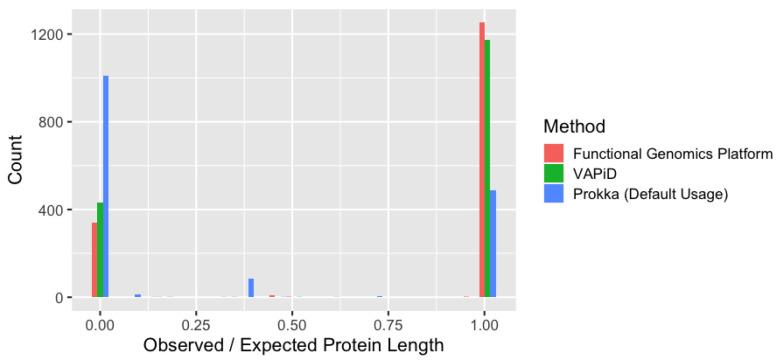
Protein length comparisons against known reference sequences for three pipelines: Functional Genomics Platform (our method), VAPiD, Prokka with default usage for viral genomes. The count of protein sequences at each observed/expected value is plotted for each pipeline. Length is set to zero if a protein is missing in the results from that pipeline but present in another.

**Figure 6 viruses-13-02426-f006:**
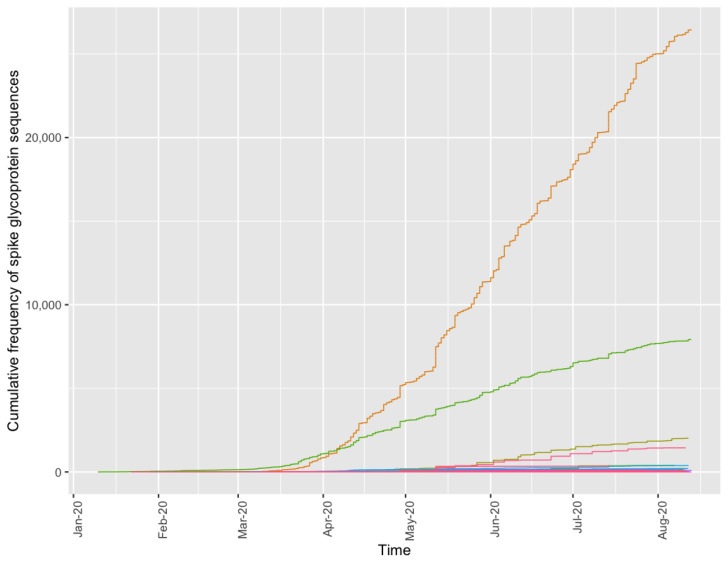
Spike glycoprotein variantsobserved in SARS-CoV-2 genomes over time and geography. Each line represents the cumulative frequency per variant (orange: D614G, green: UniProt ID P0DTC2, olive: P1140X, pink: S2 cleavage product). Low frequency S protein sequences (<5 observations) are removed from plotting for simplicity.

**Figure 7 viruses-13-02426-f007:**
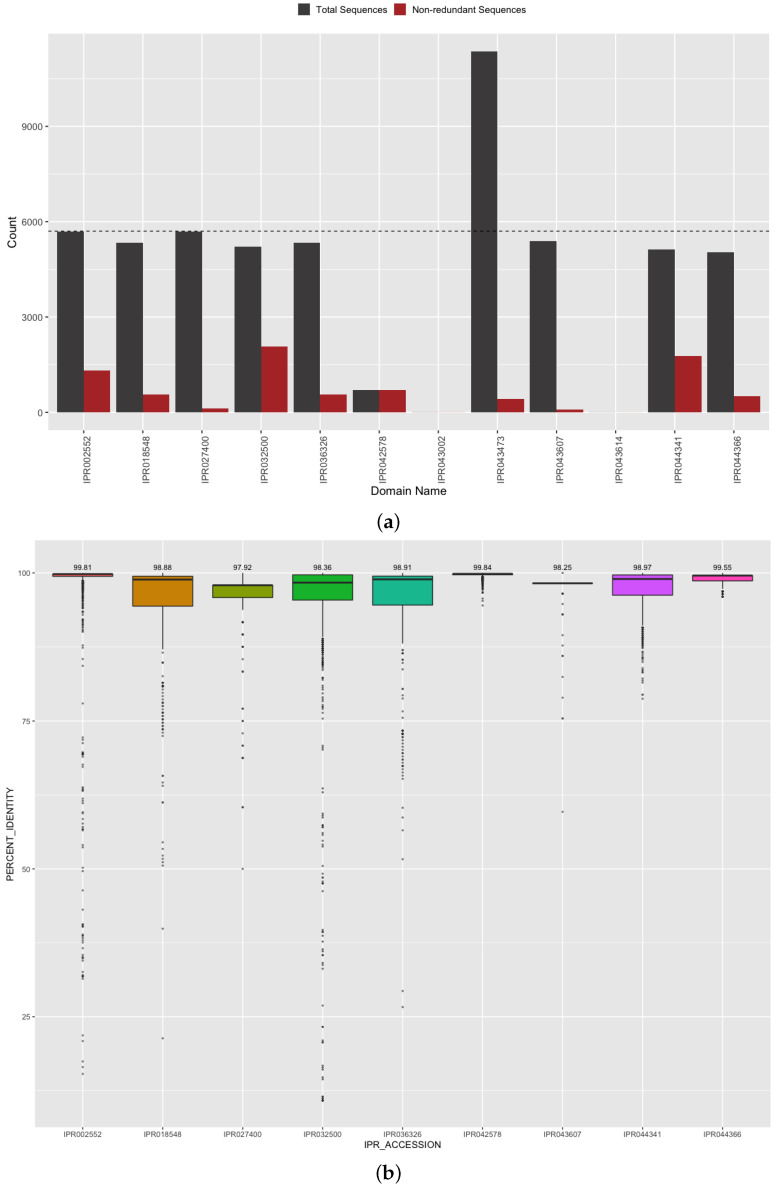
Domain annotation set membership and sequence accuracy for spike glycoprotein. (**a**) Set membership of all (grey) and distinct (red) domain sequences identified in spike glycoprotein. Count indicates the number of times a given domain was observed for all S proteins (dashed line). (**b**) The percent identity of our predicted domain sequences is calculated against reference domain sequences. For false positive domains not present in the spike protein, there is no reference sequence; therefore, these entries are omitted from this plot.

**Figure 8 viruses-13-02426-f008:**
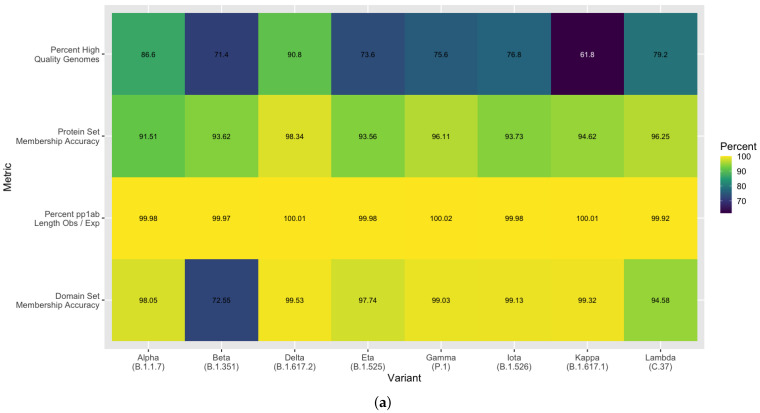
Method accuracy across a variant-diverse set of SARS-CoV-2 genomes. (**a**) Methodological evaluation was completed across eight variants of concern or interest with accuracy metrics including: genomes passing quality thresholds (percent high quality), the ability to identify all expected proteins per genome (protein set membership accuracy), length similarity between predicted pp1ab protein sequences and the reference sequence (pp1ab observed/expected), and targeted analysis to examine spike glycoprotein functional domains (domain set membership accuracy). (**b**) Expected mutations are characterized in a random sampling of genomic sequences across current variants of concern or interest. Each mutation is shown on the *y*-axis with a single asterisk indicating a mutation of interest and two asterisks indicating a mutation of concern. Grey indicates not applicable, i.e., the mutation is not expected for that variant. For the sampled genomes above quality thresholds, the percent of the mutated amino acid or deletion observed is shown.

**Table 1 viruses-13-02426-t001:** Distribution of Duplicated Genome Sequences. The number of unique genome sequences and total genome accessions (count of all entries) are listed for each source: GISAID, GenBank, and NCBI RefSeq. Pairwise comparisons are then made for each genome sequence to indicate how many times a genome sequence is duplicated within its source or between sources and for how many entries this accounts for (total accessions). For RefSeq comparisons, note that only one SARS-CoV-2 reference genome sequence is available NC_045512.2.

Source 1	Source 2	Unique Sequences	Total Accessions
GISAID	NA	47,908	55,708
GENBANK	NA	9398	11,196
REFSEQ	NA	1	1
GISAID	GISAID	2791	10,528
GISAID	GENBANK	5559	13,977
GENBANK	GENBANK	706	2504
GISAID	REFSEQ	1	43
GENBANK	REFSEQ	1	11

**Table 2 viruses-13-02426-t002:** Observed gene, protein, and functional domain biological entities in total count (redundant) and unique sequences (distinct) in high quality genomes.

Type	Total Count	Unique Count
Gene	936,603	59,531
Protein	815,878	42,611
Domain	11,621,784	59,271

## Data Availability

To support the reuse of this method and data resources, we have created multiple data access points. Pertaining to this manuscript specifically, protein and domain sequence data are provided in Appendix A, respectively with identifier mappings described in Appendix A. The data generated from the extension of this method to additional variant-diverse genome sequences can be accessed via Cloud Object Store with instructions in Appendix A. The Functional Genomics Platform is available at https://ibm.biz/functional-genomics, which provides a developer toolkit (REST services, omxware Python SDK, and Docker container) or web interface and can be accessed by requesting credentials at the link above. All the GISAID data are available at www.gisaid.org.

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
