# Peer review of "Semi-Supervised Pipeline for Autonomous Annotation of SARS-CoV-2 Genomes"

_viruses, 2021, doi:10.3390/v13122426_

Round 1

Reviewer 1 Report

Aim of the study: create an automated, autonomous SARS-Cov-2 genome annotation tool

Advantage over existing tools : use of multiple instead of a single reference sequence to annotate genomes, which could be especially relevant with the annotation of more extensively mutated sequences.

Authors modified a previously described pipeline to add SARS-CoV-2 specific annotation capability. The current manuscript describes these modifications.

Line 41: mention that the polybasic site is a furin cleavage site, ie say ….polybasic furin cleavage site…

Line 44-45: should this be: …ORF1a (responsible for pp1a) continues as ORF1b (responsible for pp1ab)…? ORF2 is the spike protein.

Related to this: how is ORF2 annotated in the new tool?

This manuscript seems more suitable for a bioinformatics-focused journal given that the content is the detailed description of a software algorithm.

Author Response

We thank the reviewer for their time and feedback. Please find the attached response to the review.

Reviewer 2 Report

  1. A) General comments

The study was interesting and will be useful for the expanding SARS-CoV-2 genome databases. However, the authors have to organize and re-structure the manuscript to fit the audiences of the Journal. I am sure most of the audiences of this journal are virologists. Although they are interested to this topic, they will not understand the details. Only a small proportion of the audiences are capable of going through this manuscript. To proceed the revision, the authors are suggested to work with colleagues of medical field. For example, try to avoid using computer terms such as ‘we computed md5 hashes’ (line 293), ‘we benchmarked our pipeline’ (line 347).

  1. B) Specific comments

- title: should be revised, my suggestion: Semi-supervised pipeline to annotate SARS-CoV-2 genome automatically

- introduction section: a table can be used to summarize the research gaps between the existing methods and your pipeline. This table should contain at least four columns: (1) limitations, (2) existing methods, (3) aims, (4) current pipeline

For example:

(1) Uses published gene coordinates to extract sequences from the query genome based on positional and sequence similarity to a reference genome

(2) NextStrain’s Augur, Bowtie2, UCSC SARS-CoV-2 genome browser

(3) To decrease the dependency on a single reference genome.

(4) Reference genome is not required

- section 2: you only need to outline the objectives for each section, you can consider to put to specific technical details in the Appendix section.

2.1 To retrieve the genomes for analysis

2.2 To annotate the genomes

2.2.1 modifications

2.2.2 modifications

2.3 Protein domain annotation

2.4 Comparison study

On the other hand, you can use a figure to illustrate the workflow to assess your pipeline.

- lines 247-253: this section can be omitted, when the genome is annotated successfully, mutations within each gene should not be incorrect.

- lines 381-437: similarly, section 3.5 can be deleted.

- line 67: you introduced a very good point, recombination. However, this parameter was not assessed in your manuscript.

Try to build different SARS-CoV-2 genomes by mixing different genes, e.g. pp1ab gene from alpha variant + S gene from delta variant.

On the other hand, instead of using SARS-CoV-2 genomes, try to mimic genomes by mixing certain genes from other coronaviruses to assess if your pipeline can annotate the genome.

It will be more interesting and can replace lines 247-253 and 381-437.

  1. C) Typos / minor comments

- discordant for the total number of SARS-CoV-2 proteins mentioned:

line 306, it mentioned fourteen protein products

Figure 3a, only 13 proteins were shown

Supplementary file 3, 13 protein names

- Figure 3a: suggest to follow the genome structure to list those 13 proteins, from 5’end to 3’end.

You can have a look to this article:

Naqvi AAT, Fatima K, Mohammad T, Fatima U, Singh IK, Singh A, Atif SM, Hariprasad G, Hasan GM, Hassan MI. Insights into SARS-CoV-2 genome, structure, evolution, pathogenesis and therapies: Structural genomics approach. Biochim Biophys Acta Mol Basis Dis. 2020 Oct 1;1866(10):165878. doi: 10.1016/j.bbadis.2020.165878. Epub 2020 Jun 13. PMID: 32544429; PMCID: PMC7293463.

In addition, try to follow the protein names used.

Please note orf9b is not shown in this article.

Author Response

(The authors gave the same response as above.)

Round 2

Reviewer 2 Report

I do not have query for this version. However, the editors have to judge if this manuscript is suitable for publication in Viruses.